# Proximal Policy Distillation

**Giacomo Spigler** *g.spigler@tilburguniversity.edu*
*AI for Robotics Lab (AIR-Lab)*
*Department of Cognitive Science and Artificial Intelligence*
*Tilburg University*

**Reviewed on OpenReview:** *https://openreview.net/forum?id=WfVXe88oMh*

## Abstract

We introduce Proximal Policy Distillation (PPD), a novel policy distillation method that integrates student-driven distillation and Proximal Policy Optimization (PPO) to increase sample efficiency and to leverage the additional rewards that the student policy collects during distillation. To assess the efficacy of our method, we compare PPD with two common alternatives, student-distill and teacher-distill, over a wide range of reinforcement learning environments that include discrete actions and continuous control (ATARI, Mujoco, and Procgen). For each environment and method, we perform distillation to a set of target student neural networks that are smaller, identical (self-distillation), or larger than the teacher network. Our findings indicate that PPD improves sample efficiency and produces better student policies compared to typical policy distillation approaches. Moreover, PPD demonstrates greater robustness than alternative methods when distilling policies from imperfect demonstrations. The code for the paper is released as part of a new Python library built on top of stable-baselines3 to facilitate policy distillation: `https://github.com/spiglerg/sb3_distill`.

## 1 Introduction

Deep Reinforcement Learning (DRL) has been used to autonomously learn advanced behaviors on a wide range of challenging tasks, spanning from difficult games (Silver et al., 2017; Vinyals et al., 2019; Berner et al., 2019) to the control of complex robot bodies (Akkaya et al., 2019; Liu et al., 2022b; Haarnoja et al., 2024), and advanced generalist agents (Reed et al., 2022; Abramson et al., 2020; Octo Model Team et al., 2024).

However, optimizing the performance of DRL agents typically requires extensive trial-and-error, requiring many iterations to identify effective reward functions and appropriate neural network architectures. As such, considerable research effort has been made to improve the sample efficiency of reinforcement learning. A particularly effective approach is to leverage prior knowledge -such as that obtained from previous training runs (Agarwal et al., 2022)- through policy distillation (PD) (Rusu et al., 2016). Policy distillation, inspired by knowledge distillation (KD) in supervised learning (Hinton et al., 2015), focuses on transferring knowledge from a 'teacher' neural network to a 'student' network that usually differs in architecture or hyperparameters.

Despite the conceptual similarity between KD and PD, their use and implementation are very different. KD is mainly used for model compression, whereby functions learned by a high-capacity neural network are transferred to a smaller network. This helps to reduce inference time, which is crucial to run the final model on embedded hardware. This setting is not common in DRL, as the neural networks are typically kept small to shorten the long training times.

In contrast, PD is utilized differently in DRL. For example, a smaller model that is cheap to train can be 'reincarnated' into a larger one that is more expressive, possibly with different hyperparameters (Agarwal et al., 2022), to achieve better performance while limiting the computational resources required. Another effective use of PD in DRL is to use teacher model(s) as 'skills priors', e.g., useful component behaviors that solve sub-tasks of a problem, to make it easier for a student agent to learn to solve more complex tasks

(Merel et al., 2019; Liu et al., 2022b; Zhuang et al., 2023). In this case, PD can be used as a regularizing term to bias towards behaviors that are expected to be useful for the larger, more complex task, or to simply combine a number of single-task teachers into a single multitask agent (Rusu et al., 2016). Finally, policy distillation can be used to re-map the types of inputs that a neural network receives. This is useful for example in robotics, where policies trained on state-based (or even privileged) observations can be distilled into vision-based policies (Akkaya et al., 2019; Liu et al., 2022a).

Regarding implementation, KD typically involves adding a distillation loss to the problem-specific (dataset) loss function, to train a student network to reproduce the same output values as the teacher model. While several recent works have explored combining policy distillation with reinforcement learning objectives (Rajeswaran et al., 2017; Weihs et al., 2021; Nguyen et al., 2022; Shenfeld et al., 2023), many policy distillation methods still use only a distillation loss, training the student through supervised learning rather than in a reinforcement learning setting. However, this overlooks the valuable information that can be gained from the rewards that the student collects during distillation. Exploiting these rewards could accelerate the distillation process, potentially enable the student to outperform the teacher, and reduce the risk of overfitting to imperfect teachers.

We introduce Proximal Policy Distillation (PPD), a novel policy distillation method that combines student-driven distillation and Proximal Policy Optimization (PPO) (Schulman et al., 2017). PPD enhances PPO by incorporating a distillation loss to either perform traditional distillation, or to act as skills prior. Specifically, PPD enables the student to accelerate learning through distillation from a teacher, while potentially surpassing the teacher's performance. While PPD is designed and evaluated primarily with fully-observable students, it can also be applied in partially-observable tasks, for instance by using a student policy with memory (e.g., LSTM-based), provided that the teacher policy has access to the full environment state.

The main **contributions** of this paper are:

1. We introduce Proximal Policy Distillation (PPD), a novel policy distillation method that improves sample efficiency and final performance by combining student-driven distillation with policy updates by PPO.

2. We perform a thorough evaluation of PPD against two common methods for policy distillation, student-distill (on-policy distillation) and teacher-distill (Czarnecki et al., 2019), over a wide range of RL environments spanning discrete and continuous action spaces, and out-of-distribution generalization. We assess the performance of the three methods with smaller, identical (self-distillation), and larger student network sizes, compared to the teacher networks.

3. We analyze the robustness of PPD compared to the two baselines in a scenario with 'imperfect teachers', whose parameters are artificially corrupted to decrease their performance.

4. We release a new Python library, *sb3-distill* [1], which implements the three methods within the stable-baselines3 framework (Raffin et al., 2019) to improve access to useful policy distillation methods.

## 2 Proximal Policy Distillation

**Problem setting.** We consider a reinforcement learning setting based on the Markov Decision Process $(\mathcal{S}, \mathcal{A}, p, r, \rho_0, \gamma)$ (Sutton & Barto, 2018). An agent interacts with the environment in discrete timesteps $t = 0, \dots, T-1$ that together make up episodes. On the first timestep of each episode, the initial state of the environment is sampled from the initial state distribution $s_0 \sim \rho_0(s)$. Then, at each timestep the agent selects an action $a_t \in \mathcal{A}$ using the policy function $\pi_\theta : \mathcal{S} \to \mathcal{A}$, represented by a neural network with parameters $\theta$ that takes states $s_t \in \mathcal{S}$ as input. The environment dynamics are determined by the transition function $p : \mathcal{S} \times \mathcal{A} \to \mathcal{S}$ (i.e., $s_{t+1} \sim p(s_t, a_t)$). The agent receives rewards at each timestep depending on the previous state transition according to a reward function $r : \mathcal{S} \times \mathcal{A} \to \mathbb{R}$. The objective of the reinforcement learning agent is to find an optimal policy $\pi^\star$ that maximizes the expected sum of discounted rewards

---

[1] https://github.com/spiglerg/sb3_distill

$$\pi^{\star} = \arg\max_{\pi} \mathbb{E}_{\tau \sim \pi} \left[ \sum_{t=0}^{T-1} \gamma^t r_t | a_t \sim \pi(s_t), s_0 \sim \rho_0(s), s_{t+1} \sim p(s_t, a_t) \right]$$

In the context of policy distillation, we have access to a teacher agent that has been trained to solve a reinforcement learning task within the MDP framework. We particularly focus on teachers trained via actor-critic methods. Policy distillation then starts with a new student policy that is randomly initialized. Our objective is to transfer the knowledge from the teacher policy to the student while simultaneously training the student on a task using reinforcement learning. The new task can be the same as the teacher's (e.g., to implement reincarnating RL (Agarwal et al., 2022), or to perform model compression), or a new one (e.g., using the teacher as skills prior (Merel et al., 2019; Liu et al., 2022b; Tirumala et al., 2022)). The student should learn as efficiently as possible, and ideally be robust to imperfect teachers, so that the performance of the student can in principle be higher than the teacher's.

**Approach.** We build on the framework developed by Czarnecki et al. (2019) since in its general form it already allows for the inclusion of environment rewards during distillation. The general form of the distillation gradient in this framework is

$$\nabla_\theta L(\theta) = \mathbb{E}_q \left[ \sum_{t=1}^{|\tau|} \underbrace{-\nabla_\theta \log \pi_\theta(a_t \mid \tau_t) \widehat{R_t}}_{\text{policy-gradient term}} + \underbrace{\nabla_\theta \ell(\pi_\theta, V_{\pi_\theta}, \tau_t)}_{\text{distillation term}} \cdot \right] \tag{1}$$

Where $\theta$ are the parameters of the student policy $\pi_\theta$ that we wish to train (possibly in conjunction with its value function $V_{\pi_\theta}$), and $\tau = \{s_1, a_1, r_1, s_2, a_2, \ldots, r_{|\tau|}\}$ are trajectories sampled by interacting with the target environment using a control policy $q$. $\widehat{R_t}$ is a reward signal for timestep $t$, which can take many forms as in vanilla Policy Gradient (e.g., a sum of episode returns $\sum_{t'=1}^{|\tau|} r_t$, (discounted) sum of returns-to-go $\sum_{t'=t}^{|\tau|} r_{t'}$, the advantage function $A^\pi(s_t, a_t)$, etc). Finally, $\ell(\pi_\theta, V_{\pi_\theta}, \tau_t)$ is an auxiliary term for the chosen distillation method (for example, KL-divergence between teacher and student policies).

We focus on the setting where the *student* itself acts as the control policy, i.e., $q = \pi_\theta$, since it typically yields better performance after distillation than using the teacher. In fact, relying on the teacher for exploration biases the data toward its limited state-visitation distribution, leaving many states unexplored (Czarnecki et al., 2019). In this paper, we refer to distillation methods that use the student as the control policy as *student-driven*, and those that use the teacher as *teacher-driven.*

To obtain Proximal Policy Distillation (PPD), we extend the formulation from Eq. 1 by replacing the vanilla policy-gradient term with the Proximal Policy Optimization (PPO) surrogate (Schulman et al., 2017). This requires three modifications. (i) We improve sample efficiency by extending the original formulation to re-use data from a rollout buffer for several epochs, instead of using the sampled trajectories only once per update. As such, the control policy used to collect data for the $k + 1$ PPD update is the student policy from the previous step $q = \pi_{\theta_k}$, and we need to introduce importance-sampling weights $\rho_t(\theta) = \frac{\pi_\theta(a_t|s_t)}{\pi_{\theta_k}(a_t|s_t)}$. (ii) We introduce clipping to implement a proximality constraint to improve training stability, as in PPO-clip. (iii) We choose KL-divergence as distillation loss, which we also clip

$$\ell(\pi_\theta, V_{\pi_\theta}, \tau_t) = \lambda \, \mathrm{KL}(\pi_{\text{teacher}}(\cdot \mid s_t) \parallel \pi_\theta(\cdot \mid s_t)) \max(\rho_t(\theta), 1 - \epsilon)$$

Note that clipping the KL term is not required when using forward KL-divergence. However, simultaneous distillation from multiple teachers may benefit from using *reverse* KL-divergence, for which clipping is necessary. We thus keep the clipping term to retain a more general formulation.

Rather than write another gradient, we package these ingredients into an *implicit* optimization problem which is iteratively updated by mini-batch SGD:

$$\theta_{k+1} = \arg\max_{\theta} \mathbb{E}_{s,a\sim\pi_{\theta_k}} \left[ L_{\text{PPO}}(s,a,\theta) - \lambda \, \text{KL}(\pi_{\text{teacher}}(s)\|\pi_\theta(s)) \max\left( \frac{\pi_\theta(a\mid s)}{\pi_{\theta_k}(a\mid s)}, 1-\epsilon \right) \right]$$

$$L_{\text{PPO}}(s,a,\theta) = \min\left( \frac{\pi_\theta(a\mid s)}{\pi_{\theta_k}(a\mid s)} \hat{A}(s,a), g(\epsilon, \hat{A}(s,a)) \right), \qquad g(\epsilon, A) = \begin{cases} (1+\epsilon)A, & A \geq 0 \\ (1-\epsilon)A, & A < 0 \end{cases} \qquad (2)$$

where the proximality constraint is enforced by the hyperparameter $\epsilon$. $L_{\text{PPO}}$ is the PPO-clip loss, $\lambda$ is a hyperparameter that balances the relative strength of the PPO and distillation losses. $\pi_{\theta_k}$ is the policy from the previous PPO iteration that was used to collect the rollout buffer, $\pi_{\theta_{k+1}}$ is the resulting policy at the end of the iteration, which will be used to collect the next rollout, and $\pi_{\text{teacher}}$ is the teacher policy that we wish to use to guide learning. The critic is trained from scratch using a regression loss, as in PPO.

The full algorithm is reported in Appendix A (algorithm 1).

## 3 Empirical Evaluation

We compared PPD against two policy distillation methods: 'student-distill' and 'teacher-distill'. Student-distill (SD) is similar to 'on-policy distill' from Czarnecki et al. (2019), where policy distillation is treated as a supervised learning problem over trajectories collected using the student as sampling (control) policy. Teacher-distill (TD) is performed in the same way as student-distill, but the teacher policy is used to sample environment trajectories instead of the student's. Pseudocode of the two baseline methods is included in Appendix A algorithms 2 and 3).

Evaluation was performed on environments from Atari, Mujoco and Procgen because they span three axes that most affect performance in reinforcement learning and policy distillation: (i) state-space complexity – low-dimensional states (Mujoco) vs. high-dimensional pixel observations (Atari & Procgen); (ii) action spaces – discrete (Atari & Procgen) vs. continuous (Mujoco); (iii) out-of-distribution generalization – identical train/test environment (Atari & Mujoco) vs. procedurally generated train/test splits (Procgen). All tasks were episodic but discounted, thus differing from the strictly undiscounted formulation in (Czarnecki et al., 2019).

Namely, we used the following environments:

- A subset of 11 games from the **Atari** suite (`Atlantis`, `BeamRider`, `CrazyClimber`, `DemonAttack`, `Enduro`, `Freeway`, `MsPacman`, `Pong`, `Qbert`, `Seaquest`, and `Zaxxon`).

- 4 **Mujoco** environments (`Ant`, `HalfCheetah`, `Hopper`, and `Humanoid`).

- 4 environments from the **Procgen** bechmark (`coinrun`, `jumper`, `miner`, and `ninja`).

We first trained teacher models for each environment using PPO, repeating the process over five random seeds. Full details of the training procedure, hyperparameters, and network architectures are provided in Appendix A.2.

We then trained student models using the three policy distillation methods, PPD, SD, and TD, applied to three different student network architectures. These include a *smaller* network with approximately 25% of the teacher network's parameters, an *identical* network in a self-distillation scenario (Furlanello et al., 2018), and a *larger* network containing about 3-7 times more parameters than the corresponding teacher. We repeated all the experiments for each random seed. Exact network sizes and architectures are included in Appendix A.3.

**Sample efficiency.** We first examined the training curves during distillation to evaluate the sample efficiency of the different methods. Figure 1 shows the case of distillation to *larger* student networks. Similar figures for the other student network sizes are included in Appendix B.2.

We found that PPD typically learns faster and reaches better performance levels, except in the Mujoco environments, where student-distill showed a faster learning pace, although final performance was comparable in both methods.

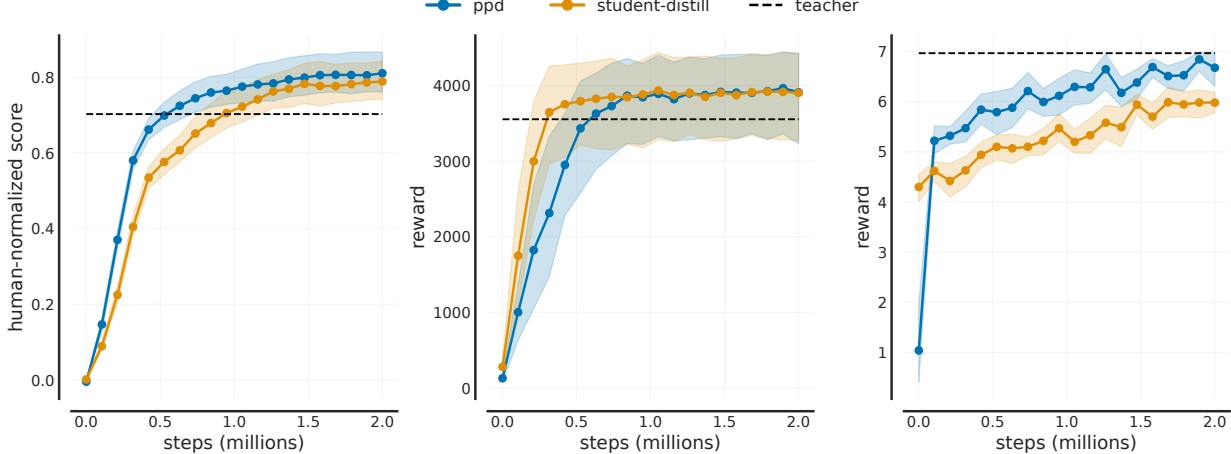

Figure 1: Training curves of PPD and student-distill in the setting of distillation onto a larger student network, showing the episodic returns collected *during* distillation. Distillation was performed over five random seeds and all environments in each of three suites 'atari', 'mujoco', and 'procgen'. The final training performance of the teacher models is shown as a dashed line. Results for procgen are calculated over training levels. We observe that PPD students achieve higher rewards than student-distill ones, and are generally more sample efficient. The teacher-distill trace is omitted here because it reflects performance on teacher-generated trajectories, leading to misleadingly high training scores; the complete curve is shown and discussed as Supplementary Figure 8. Each curve is the interquartile mean over all environments in the suite, with 95% confidence intervals.

**Distillation performance.** We then assessed the performance of student models trained with the three distillation methods. Evaluation was executed in a test setting (which in the case of 'procgen' corresponds to using a different set of levels), where the distilled students were used to interact with the environment. Actions were chosen deterministically, instead of the stochastic action selection used during training, except for 'procgen', where we observed that deterministic policies were prone to getting stuck, leading to lower performance for all agents. Results for Atari environments are reported as human-normalized scores, using base values from Badia et al. (2020).

Table 1 shows the results with respect to the corresponding teacher's performance. We found that PPD outperforms both student-distill and teacher-distill across all environments. Additionally, we observed that, like in Figure 1, the models perform close to each other on Mujoco environments.

Furthermore, we noted that distilling into larger student networks generally results in better student performance after distillation, often surpassing that of the teacher models, especially for PPD.

We further show that PPD substantially outperforms a baseline with only PPO (without distillation) in Appendix B.4.

**Distillation from imperfect teachers.** Since the goal of policy distillation is to reproduce the behavior of a teacher policy into a student network, the performance of the student is generally limited by the performance of the teacher. However, we found that PPD often achieves performance superior to its teacher. We investigated this aspect of the method further with a simple experiment where we deliberately corrupted the performance of the teacher models by perturbing their parameters with Gaussian noise

$$\theta_i \leftarrow \theta_i + \epsilon$$

Table 1: Performance of student models of three sizes (smaller, same, larger), trained using the three distillation methods (PPD, student-distill, and teacher-distill). Evaluation is performed in a test setting (which in the case of 'procgen' corresponds to using a different set of levels). Results are calculated as fraction of teacher score for each environment and random seed, and then aggregated by geometric interquartile mean (95% confidence intervals are included in brackets).

| env | smaller | | |
| --- | --- | --- | --- |
| | td | sd | ppd |
| atari | 0.86x [0.81, 0.9] | 0.94x [0.89, 1.0] | **0.97x** [0.94, 1.01] |
| mujoco | **0.99x** [0.98, 1.0] | **0.99x** [0.98, 1.0] | **0.99x** [0.97, 1.0] |
| procgen | 0.72x [0.69, 0.77] | 0.78x [0.74, 0.83] | **0.93x** [0.87, 1.01] |
| | same-size | | |
| | td | sd | ppd |
| atari | 1.0x [0.97, 1.04] | 1.07x [1.03, 1.12] | **1.08x** [1.04, 1.13] |
| mujoco | 0.99x [0.98, 1.0] | **1.0x** [0.99, 1.0] | **1.0x** [1.0, 1.02] |
| procgen | 0.84x [0.78, 0.89] | 0.88x [0.84, 0.94] | **1.03x** [0.97, 1.08] |
| | larger | | |
| | td | sd | ppd |
| atari | 1.07x [1.03, 1.13] | 1.09x [1.05, 1.15] | **1.1x** [1.05, 1.16] |
| mujoco | 0.99x [0.97, 1.0] | **1.0x** [1.0, 1.02] | **1.0x** [0.99, 1.01] |
| procgen | 0.91x [0.86, 0.98] | 0.96x [0.93, 1.0] | **1.04x** [0.98, 1.08] |

$$\epsilon \sim \mathcal{N}(0, \sigma^2)$$

with $\sigma = 0.05$. We found that it can be challenging to obtain teachers with degraded performance that still manage to perform effectively in their target environments. For this reason, we focused on a subset of the environments (four Atari environments {BeamRider, CrazyClimber, MsPacman, Qbert} and four Procgen environments {miner, jumper, coinrun, ninja}) for which reasonable imperfect teachers could be generated.

We then performed policy distillation using the three methods, and compared their performance relative to that of the original non-corrupted teachers. Evaluation was limited to students with larger networks. For Procgen, we evaluated the trained agents using test-levels (results for training levels are available in the Appendix, Table 9). Each experiment was repeated five times with different random seeds.

As shown in Table 2, PPD was found to consistently achieve better performance than both student-distill and teacher-distill, which instead fared only marginally better than their respective corrupted teachers.

Table 2: Performance of student models, trained using the three distillation methods (PPD, student-distill, and teacher-distill) using 'imperfect teachers' that are artificially corrupted to decrease in performance. Results are calculated as a fraction of the original teacher score for each environment and random seed, and then aggregated by geometric interquartile mean (95% confidence intervals are included in brackets). Distillation is performed on four Atari and four Procgen environments, and onto larger student networks.

| env | corrupted teacher score | larger | | |
| --- | --- | --- | --- | --- |
| | | td | sd | ppd |
| atari | 0.34x [0.23, 0.51] | 0.4x [0.28, 0.54] | 0.46x [0.3, 0.64] | **0.64x** [0.48, 0.8] |
| procgen | 0.65x [0.63, 0.69] | 0.63x [0.59, 0.67] | 0.64x [0.61, 0.67] | **0.71x** [0.68, 0.74] |

**Effect of hyperparameter $\lambda$.** We finally explored the impact of the hyperparameter $\lambda$, which balances the PPO and distillation losses, on distillation performance. The analysis was limited to the same subset of Atari environments as in the study of imperfect teachers, and focused on distillation onto the larger network architecture. The value of the hyperparameter was varied over four values $\lambda \in \{0.5, 1, 2, 5\}$.

Figure 2 shows that increasing the value of $\lambda$ speeds up distillation, due to the higher weight given to the distillation loss. Nevertheless, the impact of this hyperparameter is relatively small.

Test-time evaluation of the PPD students with different $\lambda$ showed that lower values allow students to learn better policies than the corresponding teacher, as expected since high $\lambda$ prioritizes the distillation loss against the PPO loss, leaving less margin for the student to deviate from the teacher's actions. The relative final test performance of students compared to the teacher are shown in parentheses in the legend of Figure 2.

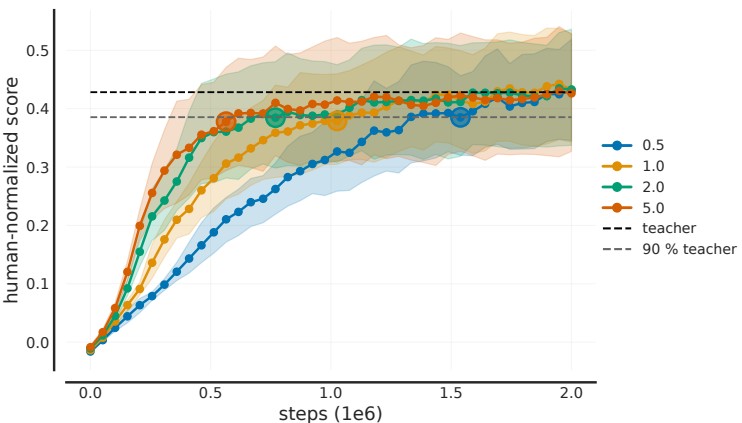

Figure 2: Performance of student models trained using PPD with different values of $\lambda \in \{0.5, 1, 2, 5\}$, over 5 random seeds. Distillation is performed on four Atari environments and onto larger student networks. Dashed lines indicate 100% (black) and 90% (gray) of the performance of the teacher models. All curves denote interquartile means with 95% confidence intervals. A circle on each curve marks the last training step where the student's performance is below 90% of the teacher's. All curves were collected during training, as in Figure 1.

## 4 Related Work

Many current policy distillation methods overlook the rewards obtained by the student during the distillation process. Instead, they frame policy distillation solely as a supervised learning task. In this approach, a dataset of state observations and teacher outputs is collected using either the student or teacher policies. The data is then stored in a distillation dataset from which training mini-batches are sampled.

A significant body of recent work focuses on distillation from teachers with privileged information or full observability to partially-observable students (Nguyen et al., 2022; Weihs et al., 2021), where pure supervised learning cannot fully capture the teacher's behavior. In these scenarios, the combination of RL with distillation becomes crucial for effective learning. While PPD was primarily designed for fully-observable settings, its integration of environment rewards with distillation makes it potentially suitable for such scenarios, as suggested by its robustness to imperfect teachers (albeit in a different context).

Meanwhile, PD as a form of supervised learning can be easily adapted to transfer functions between different types of models, such as value-based teachers (e.g., trained with DQN) and policy-based students. This proves beneficial in scenarios like reincarnating reinforcement learning (Agarwal et al., 2022), allowing continued training with alternate RL methods after distillation. For example, Green et al. (2019) showed that teachers trained using DQN can be distilled into "PPO students,", allowing for subsequent fine-tuning with Proximal Policy Optimization (PPO) (Green et al., 2019). However, unlike in our work, PPO was not used during distillation, and the trajectories for the distillation loss were collected in a teacher-driven way, with the teacher interacting with the environment instead of the student. This method closely resembles the 'teacher_distill' strategy in our baseline, which we demonstrated to be less effective than PPD.

An alternative way to perform policy distillation is through imitation learning (IL), where demonstrations from teacher models provide specific actions as hard targets, rather than using the typical soft targets (e.g., output probabilities). For example, Zhuang et al. (2023) use DAgger (Dataset Aggregation) (Ross et al., 2011) to distill five behaviors for a robot quadruped (climb, leap, crawl, tilt, run) into a single control policy, by selecting the appropriate teacher for each part of a training obstacle course. The advantage of DAgger over naive Behavior Cloning stems from its ability to query teacher actions for new states encountered by the student, which is always possible in policy distillation, rather than relying solely on teacher-collected demonstrations. The combination of IL with RL has also been explored to improve the sample efficiency of RL methods and to bias behaviors learned with RL towards natural movements (Rajeswaran et al., 2017).

Most relevant to our approach are works that combine policy distillation with reinforcement learning in different ways. COSIL (Nguyen et al., 2022) and TGRL (Shenfeld et al., 2023) incorporate the distillation loss in the reward function, while ADVISOR (Weihs et al., 2021) adds it to a policy gradient objective like in PPD, although without clipping. All three methods, however, focus exclusively on settings with privileged teachers and partially observable students, using different methods to dynamically trade off between teacher guidance and environment rewards. Similarly, (Schmitt et al., 2018) propose using Population Based Training (Jaderberg et al., 2017) to dynamically update this relative weighting. The main difference with Proximal Policy Distillation is that PPD extends PPO by applying proximal constraints to both objectives and enabling effective student-driven exploration, while these approaches rely on balancing schemes and, in the case of COSIL and TGRL, are based on different RL algorithms (SAC and IMPALA respectively). ADVISOR, like PPD, builds on PPO but differs in how it combines the objectives. Additionally, PPD is explicitly evaluated in settings with architectural differences between teacher and student, and shows robustness to imperfect teachers even without dynamic loss balancing. While these prior works focus on the specific challenge of partial observability, we evaluate PPD across a wide range of standard environments including discrete and continuous control tasks, demonstrating its effectiveness as a general policy distillation framework.

## 5 Discussion and Conclusions

We introduced Proximal Policy Distillation (PPD), a new method for policy distillation that combines reinforcement learning by PPO with a suitably constrained distillation loss.

Through a thorough evaluation over a wide range of environments, we showed that PPD achieves higher sample efficiency and better final performance after distillation, compared to two popular distillation baselines, student-distill and teacher-distill. We suggest that this is due to the reuse of samples from the rollout buffer, together with the capacity to exploit environment rewards during distillation. We also confirmed that using the student policy to collect trajectories during distillation effectively reduces overfitting to teacher demonstrations (Czarnecki et al., 2019).

In theory, the sample efficiency of teacher-distill can be improved by first collecting a dataset of trajectories using the teacher and then applying offline supervised learning. Instead, our evaluation of teacher-distill was performed online, by immediately using and then discarding teacher trajectories. We note however that reusing demonstrations collected using the teacher policy may exacerbate overfitting to the teacher. This is shown in the Appendix (Figure 8): while teacher-distill can quickly match the teacher performance during training, that is on trajectories generated by the teacher itself, its performance drops significantly when the distilled student is used to interact with the environment (as shown in Table 1).

We also found that using larger student neural networks during distillation correlates with improved performance, even surpassing that of the original teacher. Our results thus validate the advantages of 'reincarnating' reinforcement learning agents that are first trained on simpler networks to reduce training time into larger and more capable networks (Agarwal et al., 2022; Schmitt et al., 2018). However, if the student is to continue learning using any actor-critic RL method, it becomes necessary to also distill the critic of the teacher together with its policy network (actor). In the case of PPD, this is not necessary, since the student agent can learn its value function from scratch while interacting with the environment during distillation.

We further investigated the robustness of PPD compared to the two baselines in a special case where teachers were corrupted to perform suboptimally. This setting draws from the challenging, unresolved of learning

from imperfect demonstrations (Gao et al., 2018), which stems from the fact that successful replication of a teacher's behavior would include both its good and bad actions. In our tests, we found that PPD students managed to regain a large fraction, although not all, of the original uncorrupted performance in the environments tested, outperforming both student-distill and teacher-distill. Most notably, PPD students consistently obtained higher rewards than the imperfect teachers they were learning from. Further analysis of the performance of the distilled models on training versus test levels on Procgen further suggests that both student-distill and teacher-distill remain limited by the performance of the (imperfect) teachers.

Finally, we released the 'sb3-distill' [2] library to aid reproducibility of the work and facilitate the application of policy distillation methods, particularly Proximal Policy Distillation. The library implements the three methods PPD, student-distill, and teacher-distill, within the stable-baselines3 framework (Raffin et al., 2019) by introducing a new 'PolicyDistillationAlgorithm' interface to extend 'OnPolicyAlgorithm' classes.

Future work should focus on extending PPD to use teachers as skills priors, especially in the case of multitask policy distillation. To make that practical, and to further help PPD students to achieve higher performance than their teacher, it will be beneficial to implement a dynamic balancing in the trade-off between the PPO and individual teacher distillation losses, as for example done by Schmitt et al. (2018); Weihs et al. (2021); Nguyen et al. (2022); Shenfeld et al. (2023).

## 6 Acknowledgments

This research was supported by SURF grant *EINF-5635*. We gratefully acknowledge SURF (`https://www.surf.nl`) for providing access to the National Supercomputer Snellius. We also thank Murat Kirtay and Bosong Ding for their valuable support and stimulating discussions, and the anonymous reviewers for their insightful feedback, which substantially improved this manuscript.

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

# A    Supplementary Methods

## A.1    Algorithm listings and baseline methods

We include full algorithm listings for the three distillation methods compared in this work. PPD is shown in Algorithm 1, student-distill in Algorithm 2, and teacher-distill in Algorithm 3.

---

**Algorithm 1** Proximal Policy Distillation

---

**I**nput: teacher policy $\pi_{\text{teacher}}$

Initialize student policy $\pi_\theta$ and value function $V_\phi$.

**for** $k = 1, 2, \ldots$ **do**

    Collect trajectories by running the student policy $\pi_\theta$ in the environment to fill a rollout buffer $\mathcal{D}_k = \{(s_i, a_i, r_i, s_i')\}$ with $n$ environment steps.

    Compute returns $\hat{R}_i$ and then advantage estimates, $\hat{A}_i$.

    **for** epoch $= 1, 2, n_{\text{epochs}} \ldots$ **do**

      Shuffle rollout buffer.

      **for** m $= 1, 2, n_{\text{minibatches}} \ldots$ **do**

        Extract the $m$-th mini-batch $\mathcal{D}_k^m$ from $\mathcal{D}_k$.

        Update the policy and value functions by maximizing the combined objectives via gradient descent over mini-batches

$$\theta_{k+1} = \arg\max_\theta \frac{1}{|\mathcal{D}_k^m|} \sum_{(s,a) \in \mathcal{D}_k^m} L_{\text{PPO}}(s, a, \theta) + \alpha L_{\text{entropy}}(s, a, \theta) - \lambda L_{\text{PPD}}(s, a, \theta)$$

$$\phi_{k+1} = \arg\min_\phi \frac{1}{|\mathcal{D}_k^m|} \sum_{(s,a,\hat{R}) \in \mathcal{D}_k^m} \left( V_\phi(s) - \hat{R} \right)^2$$

      where

$$L_{\text{PPO}}(s, a, \theta) = \min\left( \frac{\pi_\theta(a|s)}{\pi_{\theta_k}(a|s)} \hat{A}(s, a), g(\epsilon, \hat{A}(s, a)) \right), \qquad g(\epsilon, A) = \begin{cases} (1+\epsilon)A, & A \geq 0 \\ (1-\epsilon)A, & A < 0 \end{cases}$$

$$\textcolor{blue}{L_{\text{PPD}}(s, a, \theta) = \text{KL}(\pi_{\text{teacher}}(s) \| \pi_\theta(s)) \max\left( \frac{\pi_\theta(a|s)}{\pi_{\theta_k}(a|s)}, 1 - \epsilon \right)}$$

      **end for**

    **end for**

**end for**

---

---

**Algorithm 2** Student-Distill

---

1: **Input:** teacher policy $\pi_{\text{teacher}}$ and value function $V_{\text{teacher}}$
2: Initialize student policy $\pi_\theta$ and value function $V_\phi$.
3: **for** $k = 1, 2, \ldots$ **do**
4:     Collect trajectories by running the **student** policy $\pi_\theta$ in the environment to fill a rollout buffer $\mathcal{D}_k = \{(s_i, a_i, r_i, s_i')\}$ with $n$ environment steps.
5:     Perform a step of gradient descent to obtain new parameters $\theta_{k+1}$ and $\phi k + 1$

$$\theta_{k+1} = \theta_k - \eta \frac{1}{|\mathcal{D}_k|} \sum_{(s,a) \in \mathcal{D}_k} \text{KL}(\pi_{\text{teacher}} \| \pi_\theta) - \alpha L_{\text{entropy}}(\theta, s, a)$$

$$\phi_{k+1} = \phi_k - \eta \frac{1}{|\mathcal{D}_k|} \sum_{(s,a) \in \mathcal{D}_k^m} (V_\phi(s) - V_{\text{teacher}}(s))^2$$

6: **end for**

---

**Algorithm 3** Teacher-Distill

---

1: **Input:** teacher policy $\pi_{\text{teacher}}$ and value function $V_{\text{teacher}}$
2: Initialize student policy $\pi_\theta$ and value function $V_\phi$.
3: **for** $k = 1, 2, \ldots$ **do**
4:     Collect trajectories by running the **teacher** policy $\pi_{\text{teacher}}$ in the environment to fill a rollout buffer $\mathcal{D}_k = \{(s_i, a_i, r_i, s_i')\}$ with $n$ environment steps.
5:     Perform a step of gradient descent to obtain new parameters $\theta_{k+1}$ and $\phi k + 1$

$$\theta_{k+1} = \theta_k - \eta \frac{1}{|\mathcal{D}_k|} \sum_{(s,a) \in \mathcal{D}_k} \text{KL}(\pi_{\text{teacher}} \| \pi_\theta) - \alpha L_{\text{entropy}}(\theta, s, a)$$

$$\phi_{k+1} = \phi_k - \eta \frac{1}{|\mathcal{D}_k|} \sum_{(s,a) \in \mathcal{D}_k^m} (V_\phi(s) - V_{\text{teacher}}(s))^2$$

6: **end for**

---

### A.2 Training of the teacher models

We train teacher models for each environment over five random seeds $\{100, 200, 300, 400, 500\}$. We used the following environments from Gymnasium (Towers et al., 2023) and Procgen (Cobbe et al., 2019):

- **Atari** (11 environments): `AtlantisNoFrameskip-v4`, `SeaquestNoFrameskip-v4`, `BeamRiderNoFrameskip-v4`, `EnduroNoFrameskip-v4`, `FreewayNoFrameskip-v4`, `MsPacmanNoFrameskip-v4`, `PongNoFrameskip-v4`, `QbertNoFrameskip-v4`, `ZaxxonNoFrameskip-v4`, `DemonAttackNoFrameskip-v4`, `CrazyClimberNoFrameskip-v4`.

- **Mujoco** (5 environments): `Ant-v4`, `HalfCheetah-v4`, `Hopper-v4`, `Swimmer-v4`, `Humanoid-v4`.

- **Procgen** (4 environments): `miner`, `jumper`, `ninja`, `coinrun`.

Training on Atari and Procgen environments was performed using an IMPALA-CNN architecture with $\{16, 32, 32\}$ convolutional filters, a 256-unit fully connected layer, and ReLU activations (Espeholt et al., 2018). Separate neural networks were used for policy and value networks. Training on Atari was performed using PPO for a total of 10 million environment steps, while Procgen environments were trained for 30 million steps. Atari environments were wrapped to terminate on the first life loss, without frame skipping, and using frame stacking with the 4 most recent environment frames. For Procgen, we used 200 'easy' mode unique levels during training.

Mujoco environments used a MultiLayer Perceptron (MLP) backbone with two hidden layers of 256 units and ReLU activation. Separate networks were used to represent the policy and value functions. Training was performed for 10 million environment steps.

Rewards were always normalized using a running average, and observations were normalized when training Mujoco environments. Final normalization statistics for Mujoco teachers were then fixed and reused during distillation, to guarantee that both student and teacher received the same inputs. This was not strictly required, and it was only chosen for simplicity.

The PPO hyperparameters are shown in Table 3. Simple hyperparameter tuning was initially performed, starting from parameter values suggested from the stable-baselines3 model zoo (Raffin, 2020).

Table 3: PPO hyperparameters used for training the teacher models.

| Hyperparameter | Value |
|---|---|
| n_envs | 18 |
| n_steps | 256 (Atari, swimmer, hopper), 512 (Procgen, Mujoco) |
| batch_size | 512 |
| $\gamma$ | 0.995 |
| $\lambda$ | 0.9 |
| lr | 3e-4 |
| n_epochs | 4 |
| ent_coef | 0.01 (Atari, Procgen, swimmer, hopper), 0.0 (Mujoco) |
| total env steps | 10 million (Atari and Mujoco), 30 million (Procgen) |

### A.3 Policy distillation experiments

We performed policy distillation of all teachers, across all random seeds, for all PD methods (PPD, student-distill, teacher-distill) onto three different student network sizes: **smaller**, **same** (same architecture as the student, i.e., self-distillation), and **larger**.

The smaller network for Atari and Procgen was a Nature-CNN (Mnih et al., 2015) with convolutional filters $\{32, 32, 32\}$ ($8s4, 4s2, 3s1$) and a fully connected layer of 128 units, resulting in $\sim 0.25x$ the number of parameters of the teacher. The larger networks for Atari and Procgen were IMPALA-CNNs with $\{32, 64, 64\}$

convolutional filters and 1024 units in the fully connected layer ($\sim 7.5x$ the number of parameters of the base teacher network).

The smaller network for Mujoco was an MLP with two hidden layers of sizes 128 and 64 ($\sim 0.25x$ the number of parameters of the teacher), while the larger network had two hidden layers of size 512 ($\sim 3x$ the number of parameters of the base network)

Table 4 reports the number of parameters for all student networks.

The hyperparameters of PPD related to PPO were the same as for the teacher training, except we used $\gamma = 0.999$ during distillation ($\gamma = 0.995$ for swimmer and hopper), end_coef=0, and shorter rollout trajectories (n_steps=64 for PPD, and n_steps=5 for student-distill and teacher-distill). Distillation was performed for 2 million environment steps for all methods.

Evaluation of distilled students was performed with deterministic actions for Atari and Mujoco, and stochastic actions in Procgen. Evaluation on Procgen environments was always on unseen test levels, rather than the training levels used for training the teacher and for distillation. Results from Atari games are reported with human-normalized scores, using base values from Badia et al. (2020). Rewards were averaged over 50 randomized episodes for each trained or distilled agent and random seed, for Atari and Mujoco, and 200 randomized episodes for Procgen.

Table 4: Number of parameters of the three networks used (smaller, same, larger) for each of the three domains (Atari, Mujoco, Procgen). Values in parentheses denote the relative size of each network with respect to the original teacher.

| domain | smaller | same-size | larger |
|--------|---------|-----------|--------|
| atari | 279.3k (0.25x) | 1.1M (1x) | 8.3M (7.54x) |
| mujoco | 22.7k (0.24x) | 95.8k (1x) | 322.7k (3.37x) |
| procgen | 142.5k (0.23x) | 626k (1x) | 4.6M (7.35x) |

# B Supplementary Results

## B.1 Teacher training

Training curves for all teachers, averaged over five random seeds, are shown as follows: Figure 3 (Atari), Figure 4 (Mujoco), and Figure 5 (Procgen).

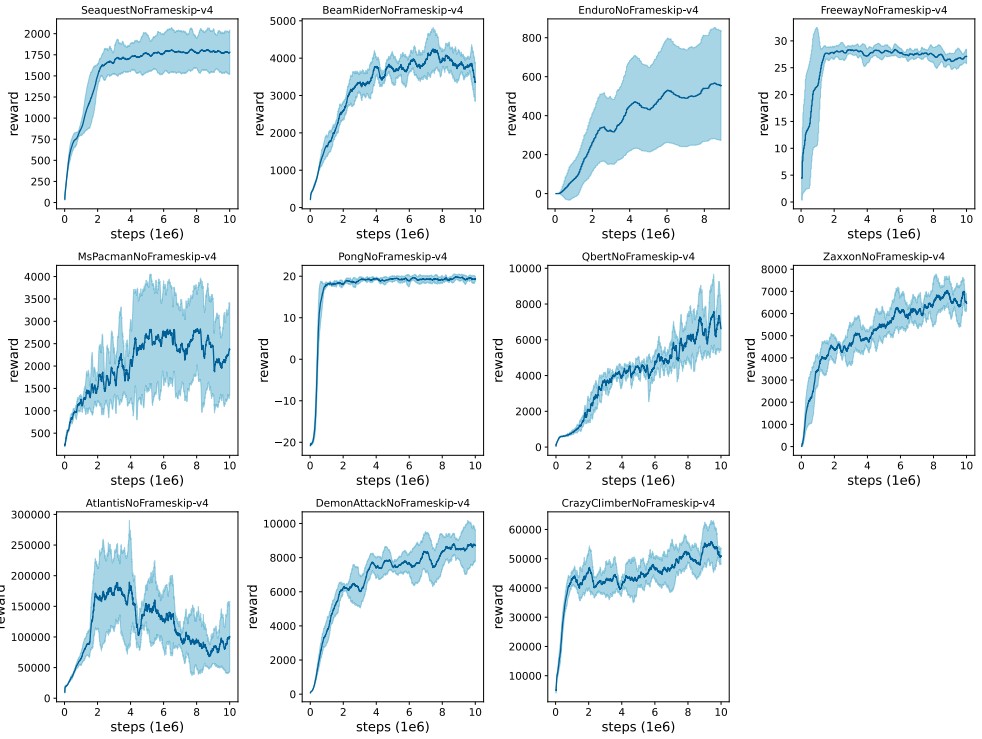

Figure 3: Training curves for all Atari teachers, averaged over 5 random seeds. Shaded areas denote standard deviation.

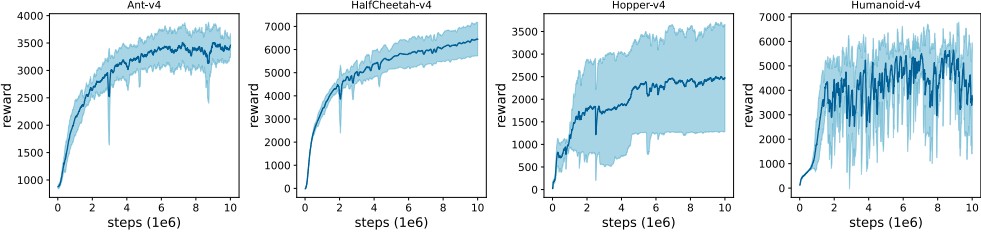

Figure 4: Training curves for all Mujoco teachers, averaged over 5 random seeds. Shaded areas denote standard deviation.

## B.2 Policy distillation

We include figures with the training performance during distillation to different sizes of student networks, extending Figure 1 from the main text. Figure 6 shows the training curves while distilling into the **smaller**

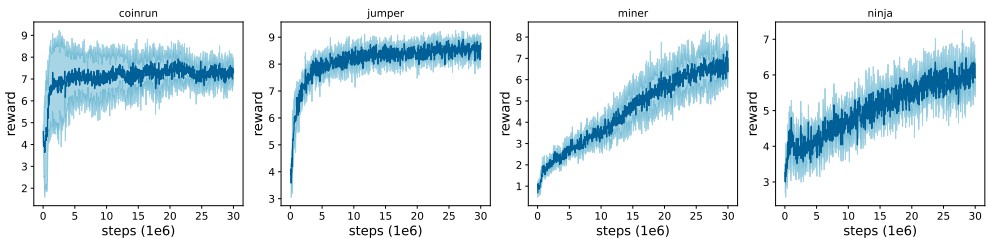

Figure 5: Training curves for all Procgen teachers, averaged over 5 random seeds. Shaded areas denote standard deviation.

student network, Figure 7 for the **same**-sized network, and Figure 8 for the **larger** network. The latter is thus like Fig. 1 from the main text, but with the inclusion of the training curve for teacher-distill.

Note that while teacher-distill approaches the performance of the teacher model quickly, the result is misleading, since performance in teacher-distill is measured over teacher trajectories. Indeed, the final performance of teacher-distill evaluated using the distilled student's trajectories (see Table 1) is worse than both other models, suggesting significant overfitting.

We then report individual scores for all environments and students in Table 5, and the individual, per-environment training trajectories for distillation to **larger** student networks for PPD and student-distill in Figures 9, 10, and 11, that is, the training curves that are combined to form Figure. 1 in the main text..

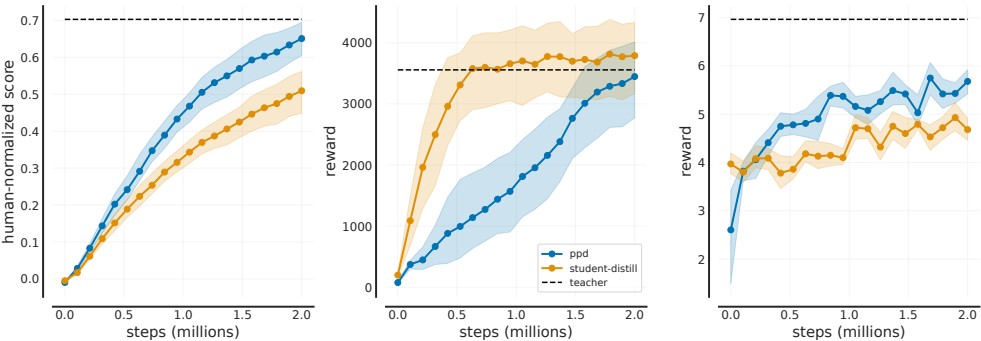

Figure 6: Same as Figure 1, but with distillation to a **smaller** student.

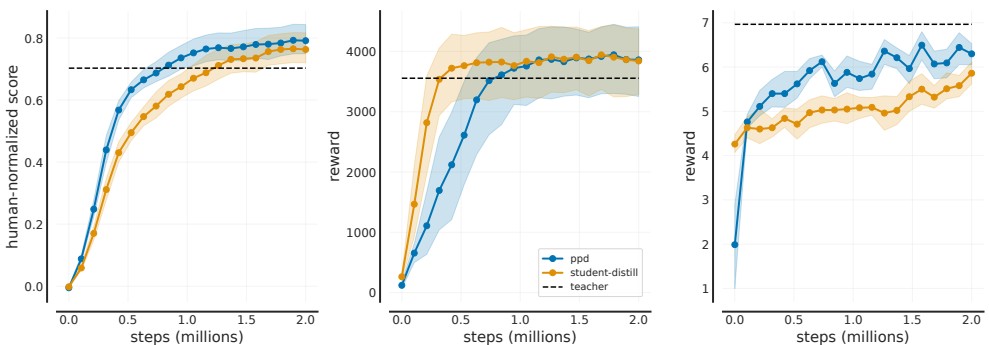

Figure 7: Same as Figure 1, but with distillation to a **same**-sized student.

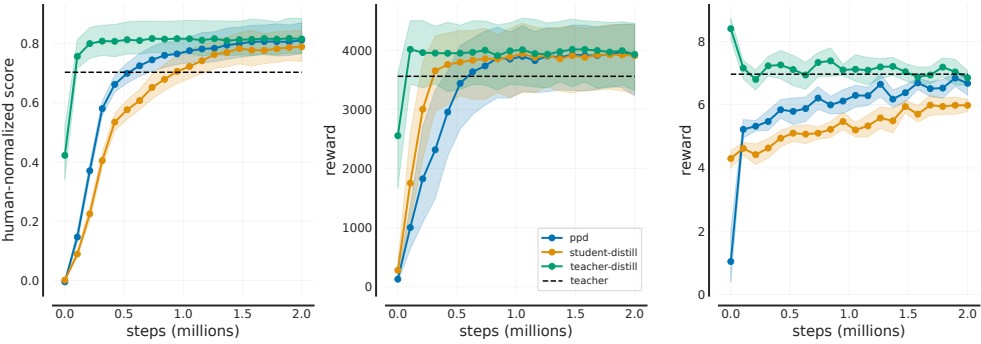

Figure 8: Same as Figure 1, showing distillation to a larger student, but also including teacher-distill. Note that while teacher-distill approaches the performance of the teacher model quickly, the result is misleading, since performance in teacher-distill is measured over teacher trajectories. Indeed, the final performance of teacher-distill evaluated using the distilled student's trajectories (see Table 1) is worse than both other models, suggesting significant overfitting.

Table 5: The table extends Table 1 from the main text by reporting results for each environment separately, averaged over 5 random seeds. Atari games are reported as human-normalized scores.

| env | teacher | smaller | | | same-size | | | larger | | |
|---|---|---|---|---|---|---|---|---|---|---|
| | score | td | sd | ppd | td | sd | ppd | td | sd | ppd |
| **atari** | | | | | | | | | | |
| atlantis | 12.85 | 21.1 | 16.27 | 23.43 | 18.07 | 14.07 | 20.42 | 18.07 | 19.64 | 28.03 |
| seaquest | 0.04 | 0.04 | 0.04 | 0.04 | 0.04 | 0.04 | 0.04 | 0.04 | 0.04 | 0.04 |
| beamrider | 0.2 | 0.12 | 0.16 | 0.2 | 0.2 | 0.23 | 0.26 | 0.22 | 0.22 | 0.25 |
| enduro | 0.69 | 0.57 | 0.61 | 0.58 | 0.79 | 0.8 | 0.88 | 0.85 | 0.85 | 0.87 |
| freeway | 0.99 | 0.73 | 0.69 | 0.98 | 0.95 | 0.97 | 1.0 | 0.98 | 0.96 | 0.99 |
| mspacman | 0.35 | 0.33 | 0.34 | 0.31 | 0.34 | 0.38 | 0.34 | 0.39 | 0.36 | 0.38 |
| pong | 1.17 | 0.98 | 1.13 | 1.14 | 1.16 | 1.17 | 1.17 | 1.17 | 1.17 | 1.17 |
| qbert | 0.58 | 0.54 | 0.58 | 0.39 | 0.66 | 0.7 | 0.73 | 0.67 | 0.76 | 0.76 |
| zaxxon | 0.69 | 0.51 | 0.59 | 0.58 | 0.57 | 0.58 | 0.68 | 0.59 | 0.6 | 0.64 |
| demonattack | 4.45 | 5.62 | 11.08 | 11.61 | 9.64 | 10.25 | 8.71 | 9.37 | 7.59 | 9.55 |
| crazyclimber | 2.08 | 1.53 | 2.83 | 2.23 | 2.19 | 2.92 | 2.33 | 2.36 | 2.35 | 2.62 |
| **mujoco** | | | | | | | | | | |
| ant | 3574 | 3529 | 3505 | 3279 | 3466 | 3543 | 3549 | 3530 | 3630 | 3490 |
| half-cheetah | 6664 | 6588 | 6530 | 6549 | 6595 | 6621 | 6638 | 6567 | 6620 | 6631 |
| hopper | 2510 | 2451 | 2521 | 2501 | 2489 | 2496 | 2534 | 2454 | 2540 | 2527 |
| humanoid | 4540 | 3747 | 4226 | 4443 | 4070 | 4591 | 4804 | 3737 | 4416 | 4618 |
| **procgen-train** | | | | | | | | | | |
| coinrun | 7.29 | 5.39 | 5.55 | 6.76 | 5.94 | 6.12 | 6.99 | 6.63 | 6.91 | 7.25 |
| jumper | 8.39 | 6.98 | 7.07 | 7.89 | 8.22 | 8.29 | 8.41 | 8.29 | 8.43 | 8.43 |
| miner | 6.64 | 1.24 | 1.47 | 1.76 | 2.34 | 2.8 | 4.39 | 3.08 | 4.11 | 5.61 |
| ninja | 6.38 | 3.44 | 4.21 | 4.61 | 4.55 | 4.73 | 5.18 | 5.23 | 5.12 | 5.65 |
| **procgen-test** | | | | | | | | | | |
| coinrun | 6.28 | 5.2 | 5.08 | 6.17 | 5.06 | 5.55 | 6.36 | 5.7 | 5.74 | 6.45 |
| jumper | 5.34 | 3.89 | 4.53 | 5.4 | 5.57 | 5.47 | 5.75 | 5.6 | 5.9 | 5.73 |
| miner | 4.57 | 1.22 | 1.46 | 1.49 | 1.87 | 2.51 | 3.66 | 2.37 | 3.34 | 4.43 |
| ninja | 4.44 | 3.34 | 3.83 | 4.33 | 4.1 | 4.13 | 4.68 | 4.29 | 4.58 | 4.57 |

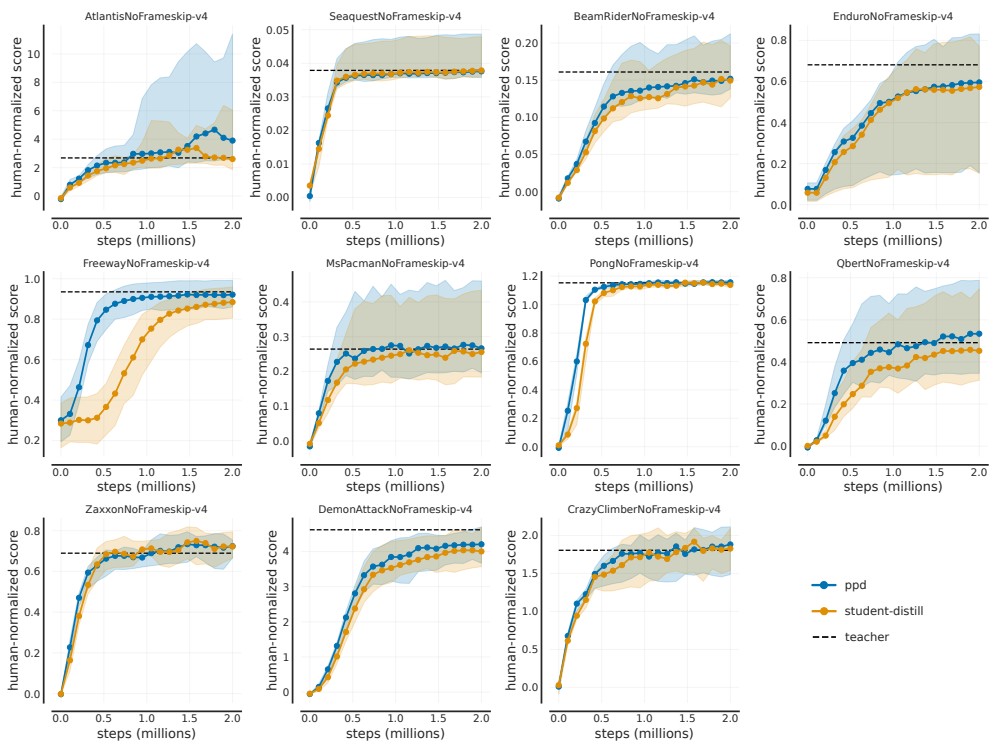

Figure 9: Individual per-environment (Atari) training curves during distillation with PPD and student-distill onto larger student networks. Figure 1 (left) combines this data obtained by interquartile averaging. Curves shown are the interquartile mean over 5 random seeds. Shaded areas denote 95% confidence intervals.

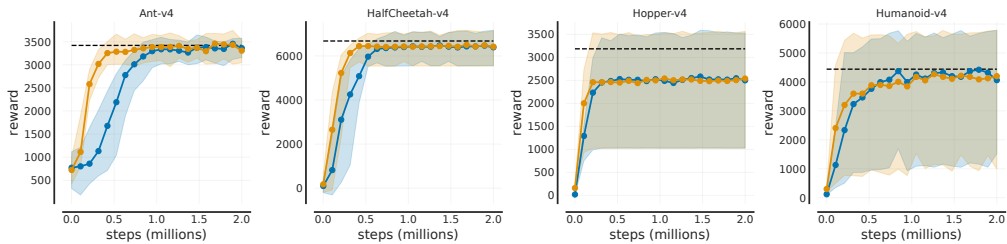

Figure 10: Individual per-environment (Mujoco) training curves during distillation with PPD and student-distill onto larger student networks. Figure 1 (left) combines this data obtained by interquartile averaging. Curves shown are the interquartile mean over 5 random seeds. Shaded areas denote 95% confidence intervals.

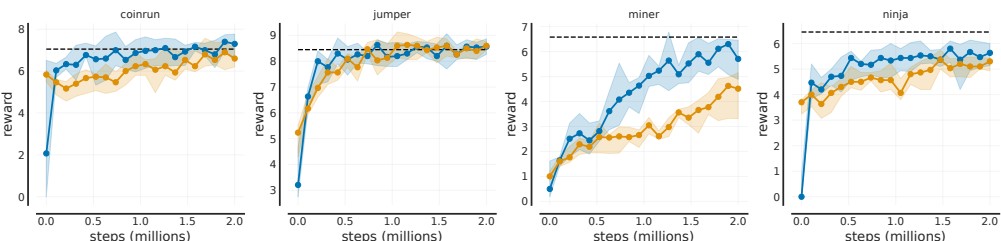

Figure 11: Individual per-environment (Procgen) training curves during distillation with PPD and student-distill onto larger student networks. Figure 1 (left) combines this data obtained by interquartile averaging. Curves shown are the interquartile mean over 5 random seeds. Shaded areas denote 95% confidence intervals.

### B.3 Distillation: imperfect teachers

We report the results of distillation from imperfect teachers for each environment in Table 6. The results from this table are aggregated and shown in Table 2 from the main text.

Table 6: Full results for each environment, extending Table 2 from the main text. We show the performance of student models, trained using the three distillation methods (PPD, student-distill, and teacher-distill) from 'imperfect teachers' that are artificially corrupted to decrease in performance. Results are calculated as a fraction of the original teacher score for each environment and random seed, and then aggregated by geometric interquartile mean (95% confidence intervals in brackets). Distillation is performed on four Atari (human-normalized scores) and four Procgen environments, and onto larger student networks.

| env | original teacher score | corrupted teacher score | td | larger sd | ppd |
|---|---|---|---|---|---|
| BeamRider | 0.2 | 0.15x | 0.14x | 0.16x | 0.19x |
| CrazyClimber | 2.08 | 0.22x | 0.22x | 0.3x | 0.41x |
| MsPacman | 0.35 | 0.38x | 0.43x | 0.47x | 0.66x |
| Qbert | 0.58 | 0.52x | 0.65x | 0.59x | 0.8x |
| atari | | 0.42x [0.33, 0.57] | 0.48x [0.38, 0.61] | 0.52x [0.38, 0.68] | **0.69x** [0.55, 0.82] |
| miner | 6.64 | 0.69x | 0.36x | 0.48x | **0.69x** |
| jumper | 8.39 | 0.56x | 0.6x | **0.63** | **0.63x** |
| coinrun | 7.29 | 0.8x | 0.71x | 0.77x | **0.83x** |
| ninja | 6.38 | 0.64x | 0.69x | 0.65x | **0.72x** |
| procgen | | 0.65x [0.63, 0.69] | 0.63x [0.59, 0.67] | 0.64x [0.61, 0.67] | **0.71x** [0.68, 0.74] |

### B.4 PPD versus PPO

We analyze the effectiveness of combining distillation and reinforcement learning in PPD by comparing against training student policies with pure PPO (without the distillation loss). We perform this comparison using larger student networks on the same subset of environments used in the imperfect teachers experiments (four Procgen and four Atari environments).

The results, shown in Table 7, demonstrate that PPD substantially outperforms pure PPO training, highlighting the value of incorporating teacher guidance through our proximal distillation approach.

Table 7: Baseline comparison of PPO (without distillation) and the three distillation methods compared in our work: teacher-distill, student-distill, and PPD. Results are calculated as fraction of teacher score for each environment and random seed, and then aggregated by geometric interquartile mean (95% confidence intervals are included in brackets).

| env | teacher score | larger | | | |
|---|---|---|---|---|---|
| | | ppo | td | sd | ppd |
| atari | 0.48 | 0.38x [0.27, 0.51] | 1.13x [1.02, 1.28] | 1.13x [1.03, 1.29] | **1.16x** [1.05, 1.33] |
| procgen (train) | 7.07 | 0.44x [0.38, 0.5] | 0.86x [0.83, 0.89] | 0.86x [0.83, 0.9] | **0.95x** [0.92, 0.98] |
| procgen (test) | 5.07 | 0.71x [0.63, 0.79] | 0.91x [0.86, 0.98] | 0.96x [0.93, 1.0] | **1.04x** [0.97, 1.08] |

### B.5 Procgen: train vs test levels

We look into the performance of teachers and students on Procgen environments, where evaluation is performed on the original **training** levels instead of the unseen test levels used for evaluation in the main text.

Table 8 extends the main results Table 1, while Table 9 shows results for the case of imperfect teachers.

Table 8: Performance of students models of three sizes (smaller, same, larger), trained using the three distillation methods (PPD, student-distill, and teacher-distill). Evaluation is performed separately on the **train** and test levels of Procgen, contrary to Table 1 that only shows performance evaluated on unseen test levels. Results are calculated as a fraction of teacher score for each environment and random seed, and then aggregated by geometric interquartile mean (95% confidence intervals in brackets).

| | teacher score | td | smaller sd | ppd |
|---|---|---|---|---|
| procgen (train) | 7.07 | 0.62x [0.6, 0.66] | 0.71x [0.66, 0.76] | **0.81x** [0.77, 0.87] |
| procgen (test) | 5.07 | 0.72x [0.69, 0.77] | 0.78x [0.74, 0.83] | **0.93x** [0.87, 1.01] |

| | teacher score | td | same-size sd | ppd |
|---|---|---|---|---|
| procgen (train) | 7.07 | 0.76x [0.74, 0.79] | 0.79x [0.75, 0.83] | **0.88x** [0.84, 0.91] |
| procgen (test) | 5.07 | 0.84x [0.78, 0.89] | 0.88x [0.84, 0.94] | **1.03x** [0.97, 1.08] |

| | teacher score | td | larger sd | ppd |
|---|---|---|---|---|
| procgen (train) | 7.07 | 0.86x [0.83, 0.89] | 0.86x [0.83, 0.9] | **0.95x** [0.92, 0.98] |
| procgen (test) | 5.07 | 0.91x [0.86, 0.98] | 0.96x [0.93, 1.0] | **1.04x** [0.98, 1.08] |

Table 9: Performance of student models, trained using the three distillation methods (PPD, student-distill, and teacher-distill) from 'imperfect teachers' that are artificially corrupted to decrease in performance. Results are calculated as a fraction of the original teacher score for each environment and random seed, and then aggregated by geometric interquartile mean (95% confidence intervals in brackets). Distillation is performed on four Atari and four Procgen environments, and onto larger student networks. Evaluation is performed on **training levels**.

| env | original score | corrupted score | td | larger sd | ppd |
|---|---|---|---|---|---|
| miner | 6.64 | 0.82x | 0.41x | 0.56x | **0.82x** |
| jumper | 8.39 | 0.89x | 0.87x | 0.88x | **0.91x** |
| coinrun | 7.29 | 0.88x | 0.86x | 0.88x | **0.92x** |
| ninja | 6.38 | 0.78x | 0.78x | 0.75x | **0.83x** |
| procgen (train) | | 0.86x [0.79, 0.91] | 0.79x [0.77, 0.83] | 0.79x [0.75, 0.83] | **0.87x** [0.84, 0.89] |

