# OpenReview forum: "Proximal Policy Distillation"
_TMLR — Accepted by TMLR_

### Review · Reviewer_AzDK · 2024-10-04

**Summary Of Contributions:**

The paper examines the problem of policy distillation in reinforcement learning. While policy distillation usually only tries to learn from a teacher, the paper suggests also performing reward maximization in cases where you have access to a reward function. By combining PPO loss and distillation loss the policy can benefit from both objectives.

**Audience:**

No

**Broader Impact Concerns:**

The paper does not contain or require a Broader Impact Statement

**Claims And Evidence:**

Yes

**Requested Changes:**

- Please update the introduction\related work with the relevant literature I mentioned above and clarify the difference between your work and theirs. This should also include Agrawal et al. 2022, which also uses a combination of RL and distillation.
- Please add 95% CI for all results in the paper. In general, I advise using the metrics proposed in the following paper as they are currently the standard in the community:
Agarwal, Rishabh, et al. "Deep reinforcement learning at the edge of the statistical precipice." Advances in neural information processing systems 34 (2021): 29304-29320.
- The proposed method outperforms the baselines in two out of three environments. Please add some analysis or discussion about why the method fails to outperform baselines in mujoco. I believe this can help in understanding the limitations of the method and be of interest to the community.
- I am missing a baseline of only performing PPO.

**Strengths And Weaknesses:**

*Strengths*
- The paper is written well and presents the motivation and the technical contributions in a clear and easy-to-follow way.
- The experimental section covers a wide array of environments, model sizes, and teachers (optimal and sub-optimal).
- Very interesting results where distillation to a model of the same size results in 10% improvement in performance. This point to the usefulness of the method as a kind of regularization, although further analysis is needed to understand it properly.

*Weaknesses*
- The paper does not acknowledge multiple prior works that also dealt with a combination of distillation loss and reward maximization:
Nguyen, Hai, et al. "Leveraging Fully Observable Policies for Learning under Partial Observability." Conference on Robot Learning. 2022.
Weihs, Luca, et al. "Bridging the imitation gap by adaptive insubordination." Advances in Neural Information Processing Systems 34 (2021): 19134-19146.
Shenfeld, Idan, et al. "Tgrl: An algorithm for teacher guided reinforcement learning." International Conference on Machine Learning. PMLR, 2023.
- Given these papers, the novelty of the proposed method is unclear and should be clarified.
- One of the differences between PPD and prior work is the clipping of the KL loss. However, the paper doesn't include an explanation or intuition of why this change is beneficial, nor ablation regarding this part. Unlike policy gradient, KL minimization does not have to be 'on-policy' and can be minimized over arbitrary sequences.

---

> ### Author Response · Authors · 2024-12-18
>
> First, we apologize for the significant delay in responding to your thoughtful review. While we have been working on implementing the suggested changes, we should have communicated our progress much sooner.
>
> We thank reviewer AzDK for the positive review and particularly appreciate making the requested changes explicit and actionable.
>
> 1) Related Work: we have thoroughly updated the introduction and related work sections to discuss the suggested papers. We have highlighted the most important changes in red in the revised manuscript. We now position PPD in relation to these works, highlighting key differences such as:
>     - Our unique use of proximal constraints on both objectives;
>     - Our broader evaluation across standard environments, compared to their focus on partially observable settings;
>     - We evaluate PPD in three important distillation regimes, namely distilling into smaller networks (model compression), larger networks (RL ‘reincarnation’), and identical networks (self-distillation; see discussion in point (3) below).
>     - PPD's effectiveness even without requiring dynamic loss balancing, which can however be easily integrated in the future, since it is independent from our method.
>
> 2) Statistical Analysis: following the suggestion by reviewer AzDK, we have updated all results to include 95% confidence intervals, using IQM as suggested in Agarwal et al. 2021. All figures and tables now reflect this improved statistical analysis.
>
> 3) Indeed, prior work on "self-distillation" (e.g., Furlanello et al. 2018) has found that knowledge distillation onto the same network architecture (in supervised learning) can improve the performance of the original teacher. An interesting theoretical analysis of self-distillation is given by Mobahi et al. 2020. However, to our knowledge, prior work on self-distillation has focused on Knowledge Distillation rather than Policy Distillation.
>
> 4) [Work in progress] MuJoCo Performance: we hypothesize that comparable performance on MuJoCo environments stems from the nature of these tasks, where achieving better performance would require fundamentally different behaviors rather than small corrections to existing policies. We would be happy to know the reviewer has any suggestions on how this could be explored further.
>
> 5) PPO baseline: as requested, we have run a baseline where PPO is used to train the student policies from scratch within the same number of environment steps, without the addition of distillation terms. The results are included in Appendix B.4 “PPD versus PPO”.

---

### Review · Reviewer_Ffw1 · 2024-10-21

**Summary Of Contributions:**

The authors propose a policy distillation algorithm that combines the PPO RL objective and a clipped distillation term.
They benchmark it on some Atari, mujoco, and procgen tasks, and show gains over pure distillation approaches that don't optimize with RL rewards. They also show their method deals with imperfect teacher policies better than pure distillation objectives, because the RL objective rewards the policy for optimal behavior.

**Audience:**

Yes

**Broader Impact Concerns:**

None.

**Claims And Evidence:**

No

**Requested Changes:**

See above, a more rigorous comparison is necessary for my recommendation.

**Strengths And Weaknesses:**

Strengths:
- the method is simple and intuitive. in addition to the PPO RL term, they add a PPO clipping loss between the teacher and student policy, just like how PPO does it to constrain the current policy and updated policy in its minibatch update.
- the paper is well written.
- there is some evidence that the method does well over pure policy distillation approaches.

Weaknesses:

- the method is not very novel, and the experimental comparison is not convincing enough to justify the method. many, many other methods combine RL and imitation learning / expert distillation objectives, see the references below for just a sampling.  **methodological novelty is not a dealbreaker to me** as simple methods that do well are good contributions. however, I am not convinced that this method is good due to the lack of baselines in the comparison.
- you only compare against pure distillation approaches. you should really compare against other methods that also combine RL and IL / distillation, as those are the most competitive baselines. it's common knowledge  that RL + distillation outperforms pure distillation, as seen by the long line of prior work.
- why not compare against PPO, which can be seen as an ablation / baseline that only uses the RL term instead of RL + distillation?
- missing ablations: why clip the distillation term with the PPO style clipping? it would be nice to show an ablation here to justify the choice.
- the imperfect teacher experiments aren't really interesting in their current state, as we would expect pure distillation approaches do worse compared to RL + distillation approaches. a more fair comparison would be to compare your method against other RL + IL approaches in this regime.

### References on methods that combine RL and imitation objectives:

These are just a few off the top of my head - see their related work for many more.

- Learning Complex Dexterous Manipulation with Deep Reinforcement Learning and Demonstrations
- Bridging the Imitation Gap by Adaptive Insubordination
- TGRL: An Algorithm for Teacher Guided Reinforcement Learning

---

> ### Author Response · Authors · 2024-12-18
>
> We sincerely apologize for our delayed response to your detailed review. While we have been implementing many of your suggestions over the past months, we should have provided updates on our progress much earlier.
>
> We thank reviewer Ffw1 for the thoughtful critique. We think that the following modifications have improved the quality of the paper.
>
> 1) Method Comparison & Literature: we have thoroughly updated the related work section to discuss the suggested papers and clarify PPD's differences from prior work combining RL and distillation. We have highlighted the most important changes in red in the revised manuscript. We now position PPD in relation to these works, highlighting key differences such as:
>     - Our unique use of proximal constraints on both objectives;
>     - Our broader evaluation across standard environments, compared to their focus on partially observable settings;
>     - We evaluate PPD in three important distillation regimes, namely distilling into smaller networks (model compression), larger networks (RL ‘reincarnation’), and identical networks (self-distillation).
>     - PPD's effectiveness even without requiring dynamic loss balancing, which can however be easily integrated in the future, since it is independent from our method.
>
> 2) Imperfect Teachers: while we agree that comparing against other RL+IL methods would be interesting, we believe our current experiments still provide valuable insights by:
>     - Demonstrating PPD's robustness without requiring dynamic balancing schemes;
>     - Showing it can recover performance even with corrupted teachers;
>     - Illustrating how PPD naturally handles teacher imperfection without requiring additional mechanisms or complexity.
>
> 3) PPO baseline: as requested, we have run a baseline where PPO is used to train the student policies from scratch within the same number of environment steps, without the addition of distillation terms. The results,  included in Appendix B.4 `PPD versus PPO’, demonstrate that PPD substantially outperforms pure PPO training, confirming the value of incorporating teacher guidance.
>
> Looking forward, we believe PPD's simple yet effective approach provides a strong foundation for further developments, including potential integration with dynamic balancing schemes as demonstrated in the prior work that is now discussed in the revised manuscript.

---

> > ### Comment · Reviewer_Ffw1 · 2025-05-28
> > **My concerns are somewhat addressed. Some room for improvement on high level motivation of distillation.**
> >
> > In my original review, I asked the authors to more throughly compare themselves against other works in distillation + RL. This work focuses on fully observed environments, whereas many prior work focus on a privileged teacher + partially observed student setting.
> >
> > This work has some interesting results in distilling teacher policies into networks of different sizes, which is a contribution I didn't fully appreciate originally.
> >
> > The authors also ran a PPO from scratch baseline, which predictably does worse since it does not assume access to a teacher policy to distill from at training time. However, this comparison is also somewhat unfair since the amount of training samples for PPO is likely lower than the amount of training samples needed to acquire the teacher policy.
> >
> > One limitation and area of discussion is when should one choose to use distillation? The paper does not seem to mention **how the teacher policies were acquired and the amount of resources necessary to get the teacher**. If we used the same amount of effort into just training the policy from scratch, then I think we would get a pretty competitive policy. So in what scenarios does it make sense to have access to a teacher policy / go through distillation instead of a more straightforward route like pure RL?
> >
> > I think if we assume this question is answered, or we assume access to a pre-existing teacher policy, this method makes a lot of sense. I would ask the author to carefully consider this question, and incorporate it into the paper to better motivate distillation.

---

> ### Author Response · Authors · 2025-06-05
>
> We thank reviewer Ffw12 for the thoughtful and constructive follow-up.
>
> To address the remaining points:
> 1) Details on teacher models training are included in Appendix A.2 'Training of the teacher models'.  To make the training cost more explicit, we have also added the number of environment steps used during training to Supplementary Table 3 (PPO hyperparameters): 10 million steps for Atari and Mujoco, and 30 million for Procgen (easy levels).
>
> 2) We agree that in scenarios where training from scratch is feasible and resources are not a bottleneck, it can lead to higher performance than policy distillation. However, in many practical situations, distillation offers important advantages. For example:
>
>      (A) As in 'Reincarnating RL' (Agarwal et al., 2022), it is often useful to be able to change model architecture without retraining from scratch, particularly when full agent training needs days or weeks.
>
>      (B)  Another promising application is multitask policy distillation. Distilling several task-specific teacher policies into a single student model can be more effective and efficient than training a multitask agent from scratch. This is because a multitask agent may learn some tasks better than others, and finetuning its performance on the weaker tasks (e.g., by changing hyperparameters or reward functions) would require retraining also on tasks that have already been learnt, increasing computational cost and complexity.

---

### Review · Reviewer_buAe · 2025-04-30

**Summary Of Contributions:**

This work introduces Proximal Policy Distillation (PPD) a novel method that augments Proximal Policy Optimisation (PPO) with policy distillation, to allow a student agent to also leverage additional knowledge from the direct interactions with the environment, next to the distillation process.

**Audience:**

Yes

**Broader Impact Concerns:**

There is no Broader Impact Statement section in the work, but I do no think one is required.

**Claims And Evidence:**

Yes

**Requested Changes:**

*Section 2:*

My main remark regards the clarity of *Section 2*, more precisely the presentation of the PPD approach. I propose the following updates, in order to improve the presentation and allow for a self-contained contribution:

Eq. 1 presents the overall distillation framework introduced by Czarnecki et al. (2019), but it is currently impossible to grasp, lacking sufficient supporting text to explain the meaning and the various components.

- First of all, I advise to clearly explain and define the meaning of "student-driven distillation". This would also allow to explain what the control/exploration policy is in your case: $q_\theta = \pi_\theta$. Secondly, please explain the remaining components and notations: $\mathscr{l}, \hat{R}$.
- In this text, "We then extend the formulation by incorporating elements from Proximal Policy Distillation (PPO)", I suppose you mean optimisation instead of distillation.
- Finally I find it hard to conceptually make the transition from Eq. 1 to Eq. 2. Can you clarify how exactly you instantiate the framework proposed by Czarnecki et al. (2019)?

*Section 3:*

- It is appreciated that experiments were carries out on three suites of environments, but can you also motive and characterise this choice in terms of type of state/action spaces? Do you, similar to Czarnecki et al. (2019), only consider undiscounted objectives and episodic tasks?
- In Figure 1, why are there no curves for teacher-distill? I see it is presented in Figure 8 in the appendix, so I am not sure why it is not part of the main text. Also do I understand correctly that results are averaged over all environments? It would be interesting to get insights per environment suite, for example understanding in more depth the Mujoco or procgen environments.

*Section 1:*
- Could you also clarify for which settings is PPD appropriate? You do mention in the related work that it targets fully-observable environments?

**Strengths And Weaknesses:**

Strengths:
+ novel approach unifying policy distillation and PPO
+ wide empirical evaluation wrt sample efficiency, performance under different teacher policy quality

Weaknesses:
- clarity and readability of some sections still needs improvement
- results could be analysed in more depth in order to better understand the scope and strenghts of the method

---

> ### Author Response · Authors · 2025-05-13
>
> We thank reviewer buAe for the thorough and constructive feedback.
> All changes are shown in red in the revised manuscript and are summarised below.
>
> 1) Clarity of Section 2 (presentation of PPD):
> - We have improved the description of Equation 1 and the framework by Czarnecki et al. (2019) to explain all components and notations.
>
> - We have better explained the meaning and use of “student-driven” and “teacher-driven” distillation.
>
> - We have rewritten the paragraph transitioning from Equation 1 to Equation 2 to make it more clear and easier to follow.
>
> - We have fixed the typo “Proximal Policy Distillation (PPO)” → “Proximal Policy Optimization (PPO)”.
>
>
> 2) Section 3 (Choice of benchmarks and Figure 1 – training curves):
> - Motivation of the choice of the three suites: we have expanded a paragraph in Section 3 to explain the specific choice of the three suites of environments:
> 	“Evaluation was performed on environments from Atari, Mujoco and Procgen because they span three axes that most affect performance in reinforcement learning and policy distillation: (i) state-space complexity – low-dimensional states (Mujoco) vs. high-dimensional pixel observations (Atari & Procgen); (ii) action spaces – discrete (Atari & Procgen) vs. continuous (Mujoco); (iii) out-of-distribution generalization – identical train/test environment (Atari & Mujoco) vs. procedurally generated train/test splits (Procgen). All tasks were episodic but discounted, thus differring from the strictly undiscounted formulation in (Czarnecki et al., 2019).”
>
> - Figure 1: we did not include the training curve of teacher-distill because it can be misleading and uninformative. In this case, performance during training is *measured over teacher-trajectories*, and thus very easy to overfit. Indeed, the lower final performance of teacher-distill evaluated using the distilled student’s trajectories (Table 1) suggest significant overfitting to the teacher (this is a main limitation of teacher-driven distillation methods).
> If the reviewer would still prefer the teacher-distill trace in the main plot, we are happy to replace Fig. 1 with the complete version from Suppl. Fig. 8, but we would first like to check whether this is still the case after these considerations. In the meantime, we have added a reference to Figure 8 in the caption of Figure 1: “The teacher-distill trace is omitted here because it reflects performance on teacher-generated trajectories, leading to misleadingly high training scores; the complete curve is shown and discussed as Supplementary Figure 8”.
>
> - Figure 1 (averaged vs individual per-environment curves): we have added the individual per-environment distillation curves in the appendix (Appendix B.2 “Policy distillation”, Figures 9-11). We have further modified the following in the caption of Figure 1: “Each curve is the interquartile mean over all environments in the suite, with 95% confidence intervals”.
>
> 3) Section 1 (in which settings is PPD appropriate):
> - PPD was designed with a fully-observable student in mind. However, it should be possible to use it in a partially-observable setting with minor modifications, for example using student policies with memory (e.g., LSTM-based) or by using an adaptive trade-off parameter λ.
>
> - Extending PPD to severe partial-observability or multi-teacher conflicts is an exciting future direction (see Sec. 5). In this regard, we had already added the following to the conclusion, while addressing the concerns of the other reviewers:
> 	“To make that practical, and to further help PPD students to achieve higher performance than their teacher, it will be beneficial to implement a dynamic balancing in the trade-off between the PPO and individual teacher distillation losses, as for example done by Schmitt et al. (2018); Weihs et al. (2021); Nguyen et al. (2022); Shenfeld et al. (2023).”
>
> - We have also added the following to the introduction section, just before the list of contributions:
> 	“While PPD is designed and evaluated primarily with fully-observable students, it can also be applied in partially-observable tasks, for instance by using a student policy with memory (e.g., LSTM-based), provided that the teacher policy has access to the full environment state.”

---

### Decision · Action_Editor_tK8C · 2025-06-02

**Recommendation:** Accept as is

**Additional Comments:**

For the final version, please do not forget to:

1. De-anonymise the paper, including little things like the source code URL.
2. Remove the colouring that was used to highlight changes in response to reviewers' comments.

**Audience:**

Yes

**Audience Explanation:**

It's a fundamental Reinforcement Learning contribution.

**Claims And Evidence:**

Yes

**Claims Explanation:**

The core contribution of this paper is a method that combines policy distillation (a student policy learning from an already-available teacher) with PPO. The PPO side of the combination, as it optimises for returns rather than similarity to the teacher policy, can help train a student that outperforms teachers (especially if the teacher is weak), whereas the distillation side can help speed up learning compared to using only PPO.

All reviewers unanimously agree that the paper is clear and well-written, and that the empirical results provide suitable support for the claims presented in the paper. I see no reason to deviate from this assessment.

---

> ### Author Response · Authors · 2025-06-06
>
> We thank the Action Editor and all reviewers for their helpful feedback.
>
> We have submitted the camera-ready version (de-anonymized, highlighted changes removed, link to GitHub repository).
> Code will be released on GitHub within a few days, after we finish cleaning it up and adding some documentation. :)